# Effect of Contact Area and Shape of Anode Current Collectors on Bacterial Community Structure in Microbial Fuel Cells

**DOI:** 10.3390/molecules27072245

**Published:** 2022-03-30

**Authors:** Agathe Paitier, Naoufel Haddour, Chantal Gondran, Timothy M. Vogel

**Affiliations:** 1Laboratoire Ampère, Ecole Centrale de Lyon, Université de Lyon, CNRS, UMR 5005, 36 Avenue Guy de Collongue, 69134 Ecully, France; agathe.paitier@ec-lyon.fr; 2Environmental Microbial Genomics, Laboratoire Ampère, Université de Lyon, CNRS, UMR 5005, 43 Boulevard du 11 Novembre 1918, CEDEX, 69616 Villeurbanne, France; vogel@univ-lyon1.fr; 3DCM, Université Grenoble Alpes, CNRS, 38000 Grenoble, France; chantal.gondran@univ-grenoble-alpes.fr

**Keywords:** microbial fuel cell, anodic biofilm, current collector, electrochemical impedance spectroscopy, power density, electroactive bacteria, carbon-based electrodes, titanium

## Abstract

Low electrical conductivity of carbon materials is a source of potential loss for large carbonaceous electrode surfaces of MFCs due to the long distance traveled by electrons to the collector. In this paper, different configurations of titanium current collectors were used to connect large surfaces of carbon cloth anodes. The current collectors had different distances and contact areas to the anode. For the same anode surface (490 cm^2^), increasing the contact area from 28 cm^2^ to 70 cm^2^ enhanced power output from 58 mW·m^−2^ to 107 mW·m^−2^. For the same contact area (28 cm^2^), decreasing the maximal distance of current collectors to anodes from 16.5 cm to 7.75 cm slightly increased power output from 50 mW·m^−2^ to 58 mW·m^−2^. Molecular biology characterization (qPCR and 16S rRNA gene sequencing) of anodic bacterial communities indicated that the *Geobacter* number was not correlated with power. Moreover, *Geobacter* and *Desulfuromonas* abundance increased with the drop in potential on the anode and with the presence of fermentative microorganisms. Electrochemical impedance spectroscopy (EIS) showed that biofilm resistance decreased with the abundance of electroactive bacteria. All these results showed that the electrical gradient arising from collectors shapes microbial communities. Consequently, current collectors influence the performance of carbon-based anodes for full-scale MFC applications.

## 1. Introduction

Microbial fuel cell (MFC) is a promising bioelectrochemical system that converts organic matter into electricity using bacterial biofilms as biocatalysts [1,2]. MFC received worldwide attention as an innovative treatment biotechnology providing energy while dealing with wastewater, industrial effluents, and agricultural waste [3,4,5]. However, electricity production and treatment efficiency of MFCs are still low for their implementation in real-world applications [6,7]. One of the biggest challenges facing the practical implementation of MFCs is the design of solutions for large-scale systems. Scale up of MFCs has been approached in different strategies including increasing the size of the system through larger electrodes [8,9,10,11] or through a constructal approach using multi-parallel small modules [12,13,14]. Scale up of MFCs using a large number of small modules is a more complicated and more expensive approach than MFCs constructed from large-scale modules. However, several studies indicated that MFC performances, in terms of energy production and treatment efficiency, cannot scale directly with electrode size [15]. Indeed, MFC performance has been shown to decrease rapidly when the surface area of electrodes increases. For example, Hsu et al. described a significant decrease of MFC power density from 23 mW·m^−2^ to 2 mW·m^−2^ by scaling up flat carbon anodes from 25 cm^2^ to 12 m^2^ [16]. The authors suggested that the majority of losses along the anode surface occur closest to current collectors, where the electric density passing through anode is the greatest. Indeed, carbon materials commonly used to make electrodes of the MFC (graphite, graphene, carbon felt, carbon cloth...) need metallic collectors for an electrical connection to the external circuit. Since the electrical conductivity of these carbon materials (10^4^–10^5^ S.m^−1^) is two to three orders of magnitude below that of metals used as collectors (titanium, stainless steel, copper...), large carbonaceous electrodes suffer a drop in potential distribution on the surface [17]. This potential drop increases with increasing the distance traveled by electrons to the collector, inducing power losses. Few studies on current collectors provide directions to reduce potential drop caused by large dimensions of electrodes [18,19]. A recent study reported a new approach of anode fabrication based on copper (Cu) foil coated with a thin thickness of conducting composite made of Polydimethylsiloxane (PDMS) doped with carbon nanofiber (CNF) [20]. In these CNF-PDMS-Cu anodes, Cu plays the role of the current collector and the distance traveled by the electrons (500 µm thick layer) is the same whatever the dimensions of the electrodes. Furthermore, Lui et al. reported an increase of power output of MFCs as the number of current collectors increased in carbon cloth anodes [19]. The authors recognized that current collectors were a source of potential loss and proposed plate collectors to reduce electron travel distance and contact resistance. Cheng et al. investigated the impact of flat current collectors in anodes on power loss of MFCs [17]. In this study, modeling and experimental analysis confirmed that power losses and drops in potential distribution arise from resistivity of the carbonaceous anode material and depend on current collector configurations. Nevertheless, the model proposed in this study was based on the assumption that bacteria evenly grow on the anode, while a potential drop could affect anodic biofilm development and consequently cause power loss. Indeed, the selection of ElectroActive Bacteria (EAB) during biofilm formation on the anode is a crucial step for improving the performance of MFCs. Several studies reported the influence of anode potential on the composition of the microbial community and electroactivity of anodic biofilms of MFCs [21,22,23]. Thus, the drop in the potential distribution caused by the contact area and shape of current collectors could restrict spatial development of EAB in anodic biofilms and could influence distribution of bacterial communities on electrode surfaces. To the best of our knowledge, the effect of current collector configurations on the microbial community in MFC anodic biofilms has not been investigated to date, although this is critical for developing the appropriate microbial selection strategies to maximize the performance of scaled-up MFCs. Therefore, in this study, the effect of different configurations of titanium current collectors on special distribution of bacterial community formed on anode surfaces was investigated. Titanium strips (1 × 14 cm^2^) were used as current collectors to connect carbon cloth (35 × 14 cm^2^) according to three configurations: MFC-A, MFC-B, and MFC-C (Figure 1). These three configurations were tested to study the effect of both the contact area of collectors and intercollector distance. Power output of MFCs was followed in relation to contact area and distance between current collectors. EIS measurements were conducted to assess MFC internal resistance. qPCR tests were performed at an increasing distance from collectors to quantify total bacteria number and, more specifically, Geobacter, a predominant electroactive genus in most anodic biofilms fed and inoculated from complex environmental media (e.g., wastewater) [24]. The 16S rRNA gene sequencing was used to describe the microbial community in each sample.

## 2. Results and Discussion

### 2.1. Effect of Current Collectors on Electricity Production Performance

Current outputs of the three configurations of MFCs (MFC-A, MFC-B, and MFC-C) were recorded as a function of the time during two weeks in order to follow biofilm growth on anodes (Figure 2A). Current outputs started to increase at the 5th day of the experiment and became stable after 10 days. These results indicated that growth kinetics of biofilms on anodes were almost the same for the three configurations. Therefore, area and shape of current collectors does significantly not impact the startup time of MFCs. Figure 2B shows polarization curves of the three configurations of MFCs obtained after 2 weeks of operation. The maximum power densities of MFCs are presented in the Appendix A). When comparing the maximum power densities obtained with the three configurations of MFCs (MFC-A, MFC-B, and MFC-C), increasing the contact area and decreasing the maximal distance of the current collectors enhanced power output. The contact area between anode cloth and metal collectors increased from 28 cm^2^ in MFC-A to 70 cm^2^ in MFC-C and maximal power density increased from 50 mW·m^−2^ to 107 mW·m^−2^, respectively. Moreover, when maximal distance to collector (the half interelectrode distance) was reduced from 15.5 cm in MFC-B to 7.25 cm in MFC-A, while maintaining the same contact area, the maximal power density slightly increased from 50 mW·m^−2^ to 58 mW·m^−2^, respectively. These results indicated that the effect of the contact area of current collectors on energy production was significant. These results are in good agreement with those reported by Hsu et al. [16]. It is also probable that the effect of collector shape on the power production is more significant for shorter intercollector distances, since the maximal distance to collector in MFC-C (3.12 cm) was smaller than that of MFC-A and MFC-B. Therefore, it is important to study the difference in bacterial community distribution on anode surface as function of the intercollector distance in each MFC. In any case, these results indicated that for small contact areas, a large part of the anode seems to be less useful for energy production and must deal with the probable congestion of electrons close to the collector.

### 2.2. Effect of Current Collectors on Bacterial Community of Anodic Biofilms

The distribution of the bacterial community on anode surfaces was investigated as a function of the intercollector distance in MFC anodes using qPCR tests and 16S rRNA sequencing. qPCR results did not reveal significant differences in total bacteria number of the anodic biofilms at increasing distances from the collectors (Figure 3A). Since *Geobacter* bacteria was intensively studied in anodic biofilms as anode-respiring bacteria producing high current densities, their distribution on anode surfaces of the three MFC configurations was investigated as a function of the intercollector distance (Figure 3B). Moreover, acetate, used as an organic substrate in this study, is the preferred electron donor for Geobacter bacteria that often enriches anodic biofilms in this bacterial family. Geobacter numbers in MFC-A, B, and C were not significantly different between samples at increasing distances from the collectors. However, overall, the Geobacter number was higher in the B system (1.1 × 10^7^ gene copies number) than in the other two configurations (4.5 × 10^6^ and 3.9 × 10^6^ for MFC-A and MFC-C, respectively). Geobacter abundances, as ratios of Geobacter to total bacteria, showed the same trend with distance than for Geobacter gene copies number, as total bacteria numbers were not very different with distance (Figure 3B). All Geobacter abundances were lower than 1%.

Sequencing of a part of the 16S rRNA gene from extracted DNA of the same anodic biofilm samples showed a higher relative Geobacter abundance than with qPCR (Figure 4). The lower ratio with qPCR may be due to the specific primers that do not comprehensively target all Geobacter species. Moreover, the number of SSU rRNA gene copies is different between bacteria. Geobacter is thought to have one [25], whereas some members of microbial communities can have several [26]. Nevertheless, the trend for Geobacter relative abundances was coherent with qPCR results with the highest abundance in MFC-B, which was around 11 ± 2%, while it was around 7 ± 1% and 4.5 ± 1.5% in MFC-A and MFC-C, respectively. The three configurations of MFCs (MFC-A, B, and C) did not show significant differences in the relative abundance of the dominant genus with increasing distances from collectors. These results indicated that Geobacter number was not correlated with power output. The reported relationship between Geobacter abundance and electricity production in mixed-species biofilms is not straightforward. Miyahara et al. suggested that Geobacter was responsible for electricity production, since they showed a correlation between maximal power of MFCs and Geobacteraceae protein content [27]. It also appeared in a previous study conducted in our laboratory wherein Geobacter was the predominant genus and whose growth was associated with MFC current production [2]. Besides, Lyon et al. described drastic differences in the microbial community composition of anodic biofilms in MFCs operated at different conditions but no significant change in power output [28]. These results demonstrated the ability of two different communities to produce similar power inputs.

The 16S rRNA gene sequencing showed that other electroactive bacteria, such as *Desulfuromonas*, *Arcobacter*, *Comamonas*, and *Lysinibacillus*, were present and in high abundances on the anodes of MFCs-A, B, and C (Table 1). Interestingly, despite the use of the same inoculum and substrate, the abundant genera were different between the three MFCs. For example, the three most abundant genus in MFC-B was electroactive with a cumulative relative abundance of 27.9%, while in MFC-A, which produced a similar maximal power, the cumulated electroactive genus provided an abundance of 16.3%. In addition, among the ten most abundant genera in MFC-A, six were fermentative bacteria, which could provide acetate or H_2_ to *Geobacter* [29]. In MFC-C, which had the highest maximal power and the most collectors, *Geobacter* was not the most abundant (4.1%) and several other electroactive genera were present in high abundance. This indicates once more that *Geobacter* may not be the only key electroactive genus in mixed-species biofilms and some others might be able to contribute significantly to electron transfer to the anode when conditions are favorable, i.e., a high electrical contact area to reduce resistance for charge transfer.

### 2.3. Influence of Current Collectors on the EIS Response of Mfcs

For more accurate analysis on the influence of current collectors on internal resistance of anodes, EIS measurements were performed for each MFC configuration after two weeks of operation. The internal resistance of MFC anodes includes different components describing the resistive and capacitive properties of anode/electrolyte interface. The Nyquist plots obtained from EIS measurements were fitted with the same electrical equivalent circuit composed of these components (Figure 5). A good match between the experimental points and the fitting curve was observed. The electrical equivalent components obtained are shown in Table 2. Figure 5 shows that the change of current collector configuration greatly influenced the impedance response of MFC anodes and affected all electrical equivalent components. Capacitances of the double layer (CPE_DL_) at the anode material/electrolyte interface showed different values according to collector configurations. Anode material/electrolyte interfaces act like capacitors, i.e., electrical charges can be stored electrostatically on the electrode and can be released when the electrode potential is suitable. This capacitance increases with the electroactive area of the geometrical surface. Indeed, Gu et al. reported a significant increase of carbon cloth capacitance from 9.6 F·m^−2^ to 5 540 F·m^−2^ after thermal treatment in air at 450 °C [32]. This was explained by a remarkably improved specific surface area after calcination. MFC-C exhibited the highest capacitance value (1.37 F) followed by MFC-A and MFC-B with 8.8 mF and 71 mF, respectively. The capacitances obtained for MFC-C were high because of the large contact area of the current collector, providing access to a large electroactive surface area. Assuming that the double layer capacitance is usually around 20 μF.cm^−2^ in aqueous media [33], the electroactive surface of the three MFC anodes was estimated by determining the ratio of CPE_DL_ to the value of double-layer capacitance. Thus, the electroactive surface area of MFC-C anode (6.8 m^2^) was estimated to be 19 times higher than that of MFC-B (3550 cm^2^) and 155 times higher than that of MFC-A (440 cm^2^). The electroactive surface area of MFC-C anode increased 140 fold compared with the geometrical surface, which may be explained by the high porosity of the carbon cloth material. These results indicated that increasing the number of collectors increased the recovery of electrochemical signal from a larger part of the anode surface. On the other hand, for the same number of collectors, the reduction of intercollector distance made the anode surface of MFC-A appear smaller than that of MFC-B, as the electric current conducted by only a part of the anode surface near the collector was recovered. This is also consistent with the observed anode ohmic resistances (R_ohm_) that are inversely correlated with the electrical conductance of anodes. R_ohm_ were very low for the MFC-C anode (0.14 ± 0.01 Ω) followed by the MFC-B anode (1.62 ± 0.05 Ω) and MFC-A anode (1.96 ± 0.06 Ω). The electroactive surface and R_ohm_ results obtained with MFC-A and MFC-B anodes were unexpected, since the reduction of the interelectrode distance was supposed to reduce electron travel distance and ohmic anode resistance. This could be explained by the effect of current collectors on the biofilm resistance (R_Bio_). Indeed, the MFC-A anode showed the highest R_Bio_ of 5.4 ± 0.2 Ω compared with those for MFC-B and C anodes (0.33 ± 0.01 Ω and 0.66 ± 0.02 Ω, respectively). R_Bio_ can be correlated to the abundance of electroactive genus in anodic biofilms. Indeed, the higher R_Bio_ of MFC-A anode corresponds to a lower abundance of electroactive genera of 16.3%, whereas in MFC-B and MFC-C this abundance increased to 30.2% and 36.3%, respectively. Several studies showed that electroactive genera such as Geobacter or Shewanella are able to produce nanowires, which are conductive proteins that improve biofilm conductivity, in pure or mixed-species biofilms [34,35,36,37]. It is likely that these conductive nanowires also improve the electrical conductance of anode materials. Unfortunately, whether the other electroactive genera are also involved in biofilm conductivity is not known yet, but the presence of electroactive genera seemed to be linked to a reduced R_Bio_. Although the current collector configuration of MFC-A was better than that of MFC-B to recover electrons from a large area, the lower abundance of nanowire networks made it still difficult. The higher R_ohm_ of the MFC-A anode could then explain why the measured electroactive surface area was smaller than that of MFC-B and MFC-C. Changes in the biofilm capacitance (CPE_Bio_) of MFC anodes was also noted. CPE_Bio_ of 1.42, 0.46, and 0.0079 mF were determined for MFC-A, MFC-B, and MFC-C, respectively. These results indicated a difference in physical and microbial structure of anodic biofilms. This difference can also be noticed through anodic charge-transfer resistance (R_CT_), which is directly related to the resistance of electron transfer at the electrode/electrolyte interface. The lowest R_CT_ was observed in MFC-C (0.62 Ω), followed by MFC-A (3.3 Ω) and MFC-B (14.3 Ω). These results suggested that the highest electrocatalytic activity was provided by the MFC-C anodic biofilm, while the lowest electrocatalytic activity was that of MFC-B. These results indicated once more that Geobacter may not be the only key electroactive genus in mixed-species biofilms and some others might be able to contribute significantly to electron transfer to the anode when conditions are favorable, i.e., a high electrical contact area to reduce resistance for charge transfer. However, EIS results showed that the electrical performance of MFCs depends mainly on the total internal resistance of anodes (R_internal_ = R_ohm_ + R_Bio_ + R_CT_). The R_internal_ of the MFC-C (1.42 ± 0.04 Ω) anode was 7 and 10 times lower than the R_internal_ of MFC-A (10.7 ± 0.4 Ω) and MFC-B (16.3 ± 0.6 Ω), respectively. It is important to note that anode materials with high capacitance improve power output of MFCs, as previously reported [33,38,39]. Indeed, an increase in performance has also been observed via galvanostatic discharge, taking advantage of the accumulated charges at the electrochemical double layer formed at the electrodes of the MFC during the charge/discharge cycles. Thus, current collector configuration would be an even more important design feature for power increase in MFCs operated with charge/discharge cycles.

### 2.4. Local Potential Influences Geobacter Development

The MFC-B anodic biofilm was composed of a larger number of Geobacter compared with other anodic biofims, whereas it did not produce the highest electrical power and did not have an optimal collectors set-up. To better understand what influenced Geobacter development on the anode, its abundance was compared to the local potential on the anode according to intercollector distance (Figure 6). The maximal potential drop of −0.102 V occurred for MFC-B from −0.318 V vs. SHE at the connection points to −0.42 V vs. SHE at the middle of the anode length, i.e., the point furthest from the collector. MFCs-A and C showed a lower potential drop of −0.038 V for MFC-A and −0.01 V for MFC-C due to lower distances between collectors. These results showed that a large part of the anode in MFC-B was at a low potential. That suggested that Geobacter together with Desulfuromonas could use the anode as an electron acceptor even at this potential and outcompeted other electroactive bacteria that could not use an electron acceptor in this low-potential range. Acetate oxidation occurs at −0.28 V vs. SHE [38], which is far above the local potential on the MFC-B anode. However, hydrogen oxidation occurs at −0.41 V vs. SHE, which was the minimum on MFC-B anode. Bacteria capable of using hydrogen as electron donor can develop at such low potentials. That is the case of Geobacter and some Desulfuromonas species [29,39], and that could explain their high abundance in MFC-B.

Commault et al. suggested that the anode potential can select for different strains of Geobacter that can also adjust their respiration chain to a potential drop [40]. 16S rRNA sequencing data revealed between eight and nine different Geobacter strains in anodic biofilm samples. Dominant strains among them were G. sulfurreducens and G. pickeringii. G. sulfurreducens was most abundant in the MFC-B (Figure 7), implying that this particular strain was able to grow at a low potential. In MFC-A, G. pickeringii accounted for 65% of Geobacter sequences. Both strains are closely related and belong to the G. metallireducens clade [41]. Ishii et al. described the G. metallireducens clade along with Desulfuromonas as being associated with low anode potential and the low current production of MFC-A was coherent with their observation [42]. In their study, the Geobacter species composition varied from the G. metallireducens clade in the early stages (first month) to Geobacter subsurface clade 2 in the later stages (third month). The microbial community described here would probably change with time, but our study showed that the current collectors set-up had an effect on Geobacter species composition from the early stages of MFC operation.

### 2.5. Economic and Technologie Considerations

The cost of anode materials is important for the practical application of MFCs. Table 3 compares the prices of anode materials used as anodes in the three MFC configurations. The major point of interest of this analysis was only based on the material price and did not take into account the processing costs that have to be included for a more detailed analysis. Based on the price of carbon cloth material (around 500 €.m^−2^), the cost of the same carbon cloth surface (490 cm^2^) used in each anode was around 24.5 €. Based on the cost of titanium collector (around 3500 €.m^−2^), increasing the contact area from 28 cm^2^ to 70 cm^2^ enhanced the cost of anode materials from 34.3 € for MFC-A and MFC-B to 49 € for MFC-C. Therefore, the cost normalized to the geometrical surface of the anodes increased from 700 €.m^−2^ for MFC-A and MFC-B anodes to 1000 €.m^−2^ for MFC-C. The cost per watt (CPW) of anode materials was calculated by dividing the cost of material by power density obtained with corresponding MFCs. The CPW is a simple measurement that can be used to compare the price/performance ratio of anode materials in the prospect of estimating the cost of capital [20]. Materials with a lower CPW are more interesting for scaling-up applications of MFCs. The CPW of MFC-C anodes (9.3 k€ W^−1^) was around 1.3 and 1.5 times lower than that of MFC-A and MFC-B anodes, respectively (Table 3). Although the increase in the surface of the current collectors increases the cost of the anodes, it decreases their CPW.

## 3. Materials and Methods

### 3.1. MFC Constrction and Monitoring

Four approx. 1.0 L planar single-chamber MFCs based on air-breathing cathodes were constructed from PVC frames and plates with internal dimensions of 35 × 14 × 2 cm^3^ (≈1 L) (Figure 8). The anodes consisted of 35 × 14 cm^^2^^ pieces of CCP-2M plain carbon cloth (Fuel Cell Earth, Woburn, MA, USA) heated to 450 °C for 30 min and placed in an ultrasonic bath for 20 min. This thermal treatment improves the specific surface area of carbon cloth electrodes and enhances their hydrophobicity owing to the introduction of oxygen functional groups [32]. The resistivity of the anodes was 1.7 Ω. cm. Untreated titanium strips (1 cm wide) were used as current collectors by pressed contact with the anode. Four anodic contacts patterns were tested (Figure 1). An air-cathode with the same surface area as the anode was made out of the same carbon cloth prepared as previously described [43] by applying platinum (0.25 mg cm^−^^2^ of electrode) and four diffusion layers and connected to the external circuit using two titanium strips along each long edge. Based on the internal resistance measured in a similar MFC previously operated, the external resistance was set at 8.2 Ω.

The MFCs were filled with 1L of synthetic wastewater to a final concentration of 500 mg COD/L as used by Lefebvre et al. [44] and with equivalent carbon and nutrients (see Appendix A for composition). The MFCs were inoculated with 5 g of dried sewage sludge (75% water mass/mass) from a Grand Lyon domestic wastewater treatment plant (Lyon, France). They were operated at room temperature with a recirculation loop at 15 mL/min with a peristaltic pump. This recirculation was used to improve homogeneity in the medium and avoid inoculum particles settling. The MFC voltage was monitored every 10 min using a Hewlett Packard 3456A Digital Voltmeter combined with an Agilent 34970A Data Acquisition/Switch Unit. If voltage dropped significantly before it stabilized, additional substrate was added into the MFC medium to maintain a concentration of 500 mg COD/L. The anodes were sampled when the voltage was stabilized by cutting off pieces (4 cm^^2^^) at different distances from the current collectors and gently rinsing pieces with sterile phosphate-buffered saline (PBS). After four days post-voltage-stabilization, when the biofilm was considered to have matured, substrate was added to the MFC medium to maintain the concentration at 500 mg COD/L. This was done in order to keep the system saturated in substrate and, thus, free from substrate limitation. Polarization curves were recorded using a potentiostat (Origaflex OGF01A, Origalys Electrochem, Lyon, France) with the anode as the working electrode and the cathode as both reference and auxiliary electrodes. Anode potential was scanned from the open circuit voltage to 0 V at a scanning rate of 1 mV.s^−1^ and current was recorded. Potential drop on the anode was calculated with Comsol 5.2 (Comsol France, Grenoble, France) from connection points where the local potential was assumed to be the equilibrium electrode potential measured against Ag/AgCl reference electrode. Titanium properties were taken from the Comsol material library. Electrical conductivity of the carbon cloth was set to 0.84 mS.m^−1^. This value is the mean conductivity of anodes (carbon cloth and biofilm) measured in six similar MFCs previously operated after several days of operation.

### 3.2. Electrochemical Impedance Spectroscopy (EIS)

EIS spectra for the anodes were recorded at the end of MFC operation before the polarization curve using a potentiostat (Origaflex OGF01A, Origalys Electrochem, Lyon, France) at open circuit potential, in a frequency range of 1 kHz to 50 mHz, with an AC signal of 20 mV amplitude (peak-to-peak) and 20 frequencies per decade. Measures used anode as the working electrode, cathode as the counter electrode, and an Ag/AgCl 3M KCl electrode inserted in the center of the chamber, equidistant between anode and cathode, as the reference electrode. Data were analyzed using Zview software. An electrical equivalent circuit taking into account the ohmic resistance, the biofilm, and the anode material/electrolyte interface was used to fit anode impedance data (Figure 5). Effective capacitance of processes was calculated from CPE parameters:(1)$$C=R^{\left(\frac{1}{\alpha }-1\right)}\times Q^{\frac{1}{\alpha }}$$
where *Q* and α are CPE parameters, and *R* is the resistance of the process in parallel to the CPE (R_CT_ or R_Bio_), all calculated by the fitting software ZView.

### 3.3. QPCR Assay

The 4 cm^2^ anode pieces were used for total DNA extraction using a DNA Soil Nucleospin kit (Macherey-Nagel, Düren, Germany). The sample lysis step was performed with a FastPrep bead beater system (MP Biomedicals, Illkirch-Graffenstaden, France) at a speed of 6 m/s for 20 s to detach bacteria from the anode. DNA was stored at −20 °C prior to qPCR. qPCR assays for Geobacter and all bacteria count were conducted on a Rotor-Gene 6000 (Corbett Life Science, Sydney, Australia). Each 20 μL reaction contained the following: 10 μL of SensiFAST SYBR No-ROX mix (Bioline), 0.8 μL of each primer (10 μM; Invitrogen, Waltham, MA, USA) (Appendix A), 6.4 μL H_2_O, and 2 μL template DNA. PCR reactions were subjected to the following cycling parameters: 95 °C for 2 min, then 30 cycles of 95 °C 15 s, 20 s at the annealing temperature, and 72 °C for 25 s. Each assay included triplicate reactions per DNA sample with three standards (containing 7 different concentrations ranging from 1 × 103 to 1 × 109 copies/µL). Quantitation was calculated using Rotor-Gene 6000 Series Software.

### 3.4. Gene Suencing

The variable V3 and V4 regions of the 16S rRNA gene were sequenced with the Illumina MiSeq system. The library was prepared following manufacturer’s instructions [32]. A first PCR amplified the variable region with the forward and reverse primers (Table 1). Each 25-µL reaction contained 22.5 µL Platinum PCR SuperMix (Thermo Fisher scientific, Waltham, MA, USA), 1 µL of 10-µM forward and reverse primers mix (Thermo Fisher scientific) and 1.5 µL DNA. PCR reactions were subjected to the following cycling parameters: 95 °C for 3 min, then 25 cycles of 95 °C for 30 s, 55 °C for 30 s, and 72 °C for 30 s and a last step at 72 °C for 5 min. Taxonomic assignation was carried out by the on-system MiSeq Reporter software based on the Greengenes database [45].

## 4. Conclusions

The present work described the impact of current collector design on overall power production of MFCs through anodic potential drop and its influence on bacterial development should be taken into account in MFC scale-up. Increasing the current collector number improved maximal power density. *Geobacter* was not correlated to power production, but its abundance seemed to be associated with fermentative microorganism presence and low anode local potential. Biofilm resistance decreased with electroactive bacteria abundance. Low anode potential favored the development of the *Geobacter* species from *G. metallireducens clade*. The double layer capacitance of the anode increased with current collectors and it generated a non-negligible capacitive current which was able to raise overall power during charge and discharge. Thus, current collectors would be an even more important design feature for power increase in MFCs operated with charge/discharge cycles. Significant additional work is needed to improve the knowledge of the mechanisms involved in conductivity and electron transfer in biofilm, and to include economic consideration and find a balance between power output improvement and cost (due to current collector material or used quantity). Further evaluation is needed for large-scale MFC applications and a range of critical challenges remain, particularly the longevity of anodic biofilms in real and long-term operating conditions.

## Figures and Tables

**Figure 1 molecules-27-02245-f001:**
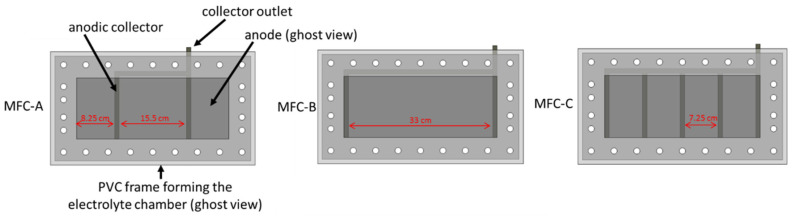
The three MFC configurations with different anodic contact patterns. Dark grey stripes represent titanium electrical contacts.

**Figure 2 molecules-27-02245-f002:**
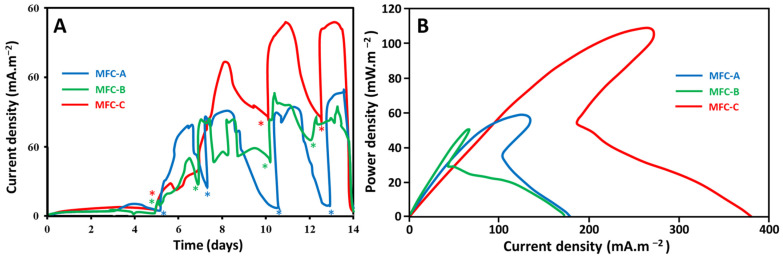
(**A**)**:** Current output as a function of time of the three configurations of MFCs. Acetate additions are indicated on the curves with (*). (**B**): Polarization curves of MFCs.

**Figure 3 molecules-27-02245-f003:**
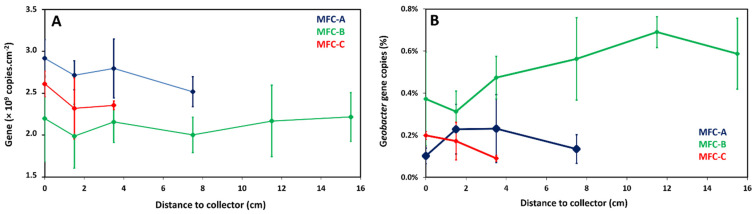
Distribution as a function of the distance from the collector of all bacteria (quantified by qPCR as gene copies number per anode cm^2^) (**A**) and of relative abundance of *Geobacter* (determined by qPCR ratios of *Geobacter* to total bacteria) (**B**).

**Figure 4 molecules-27-02245-f004:**
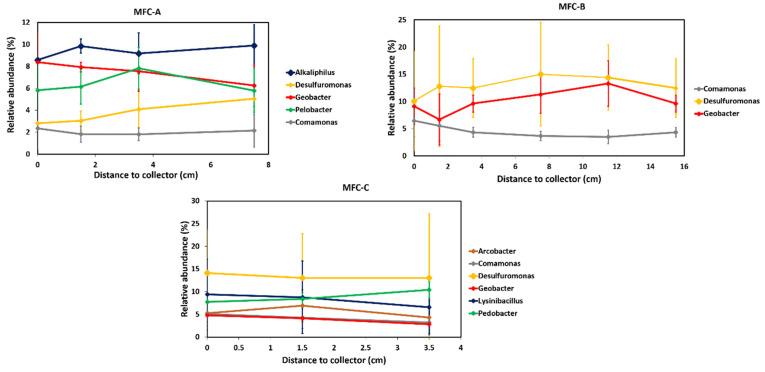
Relative abundance of genus of interest (highest abundances) in MFC-A, MFC-B, and MFC-C.

**Figure 5 molecules-27-02245-f005:**
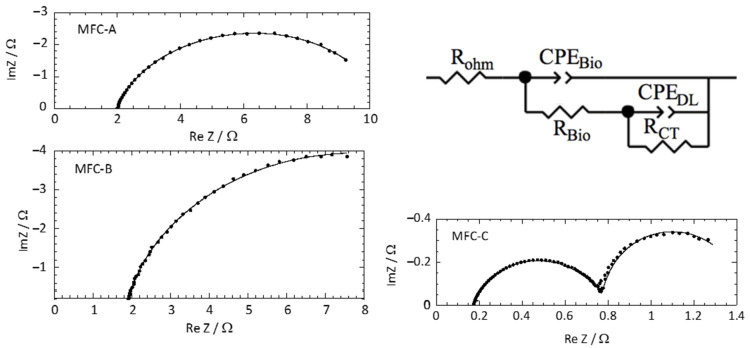
Nyquist plots for MFC-A, MFC-B, and MFC-C. Experimental (dots) and fitted data (lines) are presented. Electrical equivalent circuit used to fit anode impedance spectra. R_ohm_: electrolyte resistance; R_CT_: charge transfer resistance; R_Bio_: biofilm resistance; CPE_DL_ and CPE_Bio_: constant phase elements respectively from double layer and biofilm.

**Figure 6 molecules-27-02245-f006:**
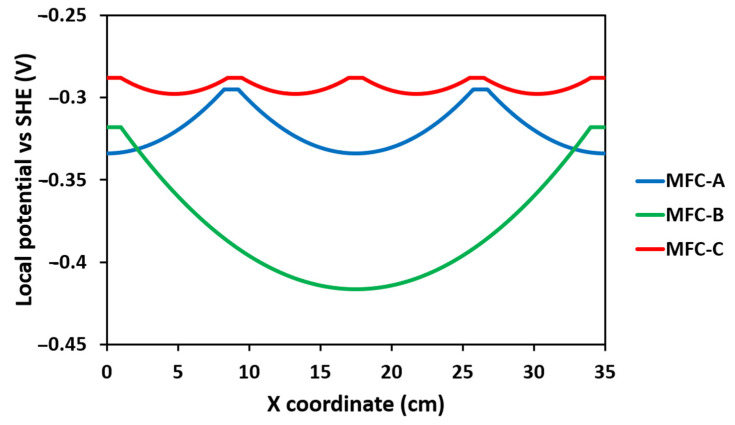
Local potential on the anode surface along the length of MFC-A, MFC-B, MFC-C.

**Figure 7 molecules-27-02245-f007:**
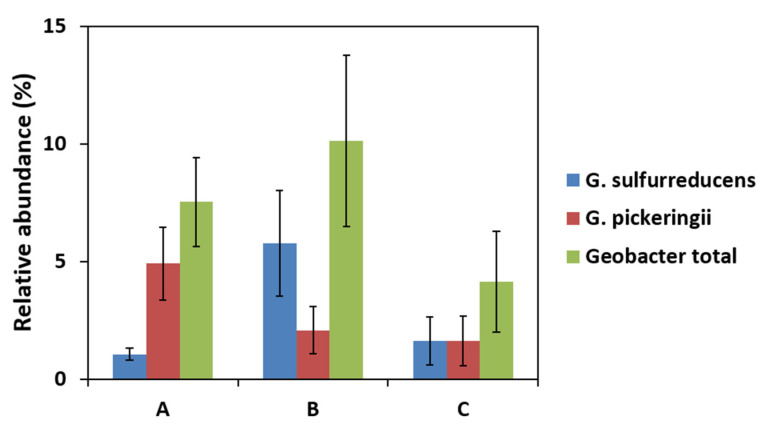
Relative abundance of genera of interest (highest abundances) in the two main Geobacter strains and total abundance of Geobacter in anodic biofilm of MFC-A (A), MFC-B (B), and MFC-C (C).

**Figure 8 molecules-27-02245-f008:**
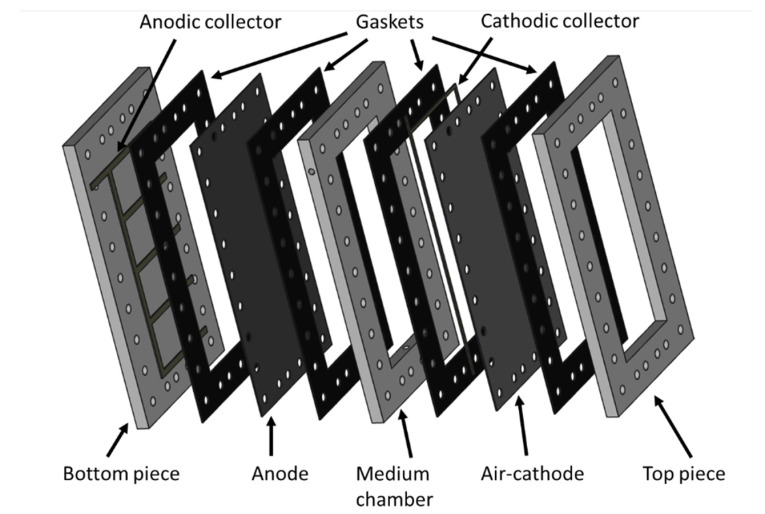
Schematic representation of the MFC reactor used in this study (here, the example of the MFC-C reactor is represented).

**Table 1 molecules-27-02245-t001:** Top 10 most abundant genera in each MFC, based on the average of relative abundance in all samples.

A		B		C	
Genus	Relative Abundance Average (%)	Genus	Relative Abundance Average (%)	Genus	Relative Abundance Average (%)
*Alkaliphilus* *	9.37	* Desulfuromonas *	13.20	* Desulfuromonas *	13.46
* Geobacter *	7.53	* Geobacter *	10.13	* Lysinibacillus *	8.59
*Pelobacter* *^,^**	6.39	* Comamonas *	4.57	*Pedobacter*	8.53
*Pedobacter*	5.46	*Sedimentibacter*	4.03	* Arcobacter *	5.74
*Parabacteroides* *	5.02	*Alkaliphilus* *	3.21	* Comamonas *	4.38
*Sedimentibacter* *	4.90	*Pedobacter*	2.98	* Geobacter *	4.14
*Propionigenium* *	4.04	*Xenophilus*	2.68	*Sedimentibacter* *	3.42
* Desulfuromonas *	3.75	*Alicycliphilus*	2.58	*Alkaliphilus* *	3.26
* Arcobacter *	2.66	*Parabacteroides* *	2.40	*Parabacteroides* *	3.14
*Clostridium* *^,^**	2.37	*Clostridium **^,^**	2.37	*Olivibacter*	2.54

*Fermenter-producing acetate *. Fermenter-producing H_2_ **. Electroactive.**Pedobacter and Olivibacter are aerobic bacteria* [30,31], * whereas anodic biofilm is supposed to be anaerobic and they have been identified as contaminants in typical DNA extraction kits and thus can be found erroneously in sequencing data. Despite their high abundance, they will not be taken into account in the following.*

**Table 2 molecules-27-02245-t002:** Ohmic resistance Rohm, biofilm resistance RBio and capacitance CBio, charge transfer resistance RCT, double-layer capacitance CDL, total internal resistance Rint measured by EIS and electroactive surface area for MFC-A, MFC-B, and MFC-C.

	MFC-A	MFC-B	MFC-C
R_ohm_ (Ω)	1.96 ± 0.06	1.62 ± 0.05	0.140 ± 0.005
R_Bio_ (Ω)	5.4 ± 0.2	0.33 ± 0.01	0.66 ± 0.02
CPE_Bio_ (F)	(1.42 ± 0.05) × 10^−3^	(4.6 ± 0.2) × 10^−4^	(7.9 ± 0.3) × 10^−6^
R_CT_ (Ω)	3.3 ± 0.1	14.3 ± 0.5	0.62 ± 0.01
CPE_DL_ (F)	(8.8 ± 0.3) × 10^−3^	0.071 ± 0.003	1.37 ± 0.4
R_internal_ = R_ohm_ + R_Bio_+ R_CT_ (Ω)	10.7 ± 0.4	16.3 ± 0.6	1.42 ± 0.04
Electroactive surface area (m^2^)	4.4 × 10^−2^	0.355	6.8

**Table 3 molecules-27-02245-t003:** Characteristics of flat materials used as anodes in MFCs.

Anode Materials	Maximum Power Density (mW·m^−2^)	Unite Price (€·m^−2^)	Price per Watt (k€·W^−1^)
MFC-A	100	700 ^a^	3
MFC-B	104	700 ^a^	16.3
MFC-C	50	1000 ^a^	10

^a^: 2021 values from Fuel Cell Earth (USA) and Goodfellow (France).

## Data Availability

Not applicable.

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
