# Peer review of "Effect of Contact Area and Shape of Anode Current Collectors on Bacterial Community Structure in Microbial Fuel Cells"

_molecules, 2022, doi:10.3390/molecules27072245_

Round 1
Reviewer 1 Report
This manuscript “effect of contact area and shape of anode current collectors on bacterial community structure in microbial fuel cells” reports that the optimization of contact area and shape of anode current collector could increase electrical gradient. However, this result is not satisfactory. The contact area increased from 28 to 70 cm2, and the power output only increased from 58 to 107 mW m-2. The power output (50 → 58 mW m-1) increased very little by the large decreased distance (16.5 to 7.75 cm). The increased contact area also brings many problems in practical application, such as poor contact, corrosion. In my opinion, this paper lacks scientific and novelty, and cannot be accepted in our journal.Author Response
Dear Colleague,
Firstly, we would like to thank you for your thoughtful review of the manuscript. We have revised our manuscript accordingly. The changes made are highlighted in the revised version of the manuscript using red colour text. Please find below your comments in bold black text and our answers in blue one.
Best regards,
Naoufel HADDOUR

Reviewer 2 Report
The manuscript reported the various configurations of Ti current collectors used to connect large surfaces of carbon cloth anodes. The current collectors had different distances and contact areas to the anode. EIS results showed that biofilm resistance decreased with increasing electroactive bacteria abundance. It can be concluded that the electrical gradient arising from the collector surface and inter-collector distance shaped the microbial communities. Hence, the current collector location and size may influence the performance of carbon-based anodes for MFC applications.
I consider the content of this manuscript will definitely meet the reading interests of the readers of the Molecules journal. However, the discussion and explanation should be further improved. Therefore, I suggest giving a minor revision and the authors need to clarify some issues. This could be a comprehensive and meaningful work after revision.
Detailed comments can be found in the PDF file.

Author Response
Dear Colleague,
Dear Colleague,
Firstly, we would like to thank you for your thoughtful review of the manuscript. They raised important issues and their inputs were very helpful for improving the manuscript. We agree with all your comments and we have revised our manuscript accordingly. The changes made are highlighted in the revised version of the manuscript using red colour text. Please find below your comments in bold black text and our answers in blue one.
Best regards,
Naoufel HADDOUR

Reviewer 3 Report
The authors have reported the effects of contact area and shape of Anode Current Collectors on Bacterial Community Structure in Microbial Fuel Cells. The manuscript narration is good, well-researched and of sound interest to the readers. However, the reviewer has a few questions regarding the research design and some of the experiments, which are as follows:
(1) The introduction is well constructed but at one point it was claimed, "the effect of current collector configurations on the microbial community in MFC anodic biofilms has not been investigated to date". The reviewer would like to point out a few reported literatures that more or less have already dealt with this subject in a more advanced way.
https://doi.org/10.1016/j.chemosphere.2018.06.070
http://dx.doi.org/10.1016/j.bios.2015.12.077
The reviewer would like to know what makes this particular study stand out.
(2) The configurations of the three control MFCs described in Figure 1 is not very clear. In fact, the line "....15.5 cm in MFC-B to 7.25 cm in MFC-A,..." (line 114, page 3) doesn't correspond to the markings in Figure 1. The authors are requested to review the line and write it in a more understandable way.
(3) There are two figure 1 in the text. Please check and correct.
(4) Figure 2 is a calibration plot, that measures the effect of the cell configuration on the output power density. It would be more appropriate to include raw data (polarisation data) on how the power densities were measured, along with this calibration plot.
(5) The impedance plot for MFC-C anode is clearly diferent from the other two anodes. Are you sure that they follow the same equivalent circuit?
(6) The voltage profile of the differently configured bioanode films during the inoculation period may be a very important parameter to report as to understand the stability at the optimum resistence operation for future applications.
(7) Some other electrochemical characterizations, like CV and/or, IV characterizations are suggested to support the data, obtained from impedance analysis.
(8) Apart from identifying the dominant electroactive bacteria genera responsible for the power output and calculating it, was there any attempt by the reviewer to study the stability of the final MFC for their practical application?
Author Response

(The authors gave the same response as above.)

Round 2
Reviewer 1 Report
I agree to accept.